# FP64 is All You Need: Rethinking Failure Modes in Physics-Informed Neural Networks

**Chenhui Xu**    **Dancheng Liu**    **Amir Nassereldine**    **Jinjun Xiong***
University at Buffalo, SUNY
Buffalo, NY, USA, 14226
{cxu26,jinjun}@buffalo.edu

## Abstract

Physics-Informed Neural Networks (PINNs) often exhibit "failure modes" in which the PDE residual loss converges while the solution error stays large, a phenomenon traditionally blamed on local optima separated from the true solution by steep loss barriers. We challenge this understanding by demonstrate that the real culprit is insufficient arithmetic precision: with standard FP32, the L-BFGS optimizer prematurely satisfies its convergence test, freezing the network in a spurious failure phase. Simply upgrading to FP64 rescues optimization, enabling vanilla PINNs to solve PDEs without any failure modes. These results reframe PINN failure modes as precision-induced stalls rather than inescapable local minima and expose a three-stage training dynamic—un-converged, failure, success—whose boundaries shift with numerical precision. Our findings emphasize that rigorous arithmetic precision is the key to dependable PDE solving with neural networks. Our code is available at `https://github.com/miniHuiHui/PINN_FP64`.

## 1 Introduction

Physics-Informed Neural Networks (PINNs) [27] have gained wide attention and applications in recent years as a novel numerical solver for partial differential equations (PDEs). PINNs can find a numerical solution by optimizing the residual loss defined by the PDE, leveraging the nature of the universal approximation [16] of neural networks and the automatic differentiation [4] provided by the deep learning framework such PyTorch [26]. Although theoretically it is capable of providing exact numerical solutions for PDEs, the researchers have identified several *failure modes* for PINNs [20]. In these cases presenting a failure mode, the PDE residual loss is optimized to a very small value, but the numerical solution provided by PINN is over trivial and has a huge error with the true solution.

To mitigate these failure modes, various methods based on optimization [29, 28], regional gradient [33, 34], sampling [12, 32, 9], and model architectures [35, 36, 7, 23] have been proposed. These methods are based on a consensual understanding of the PINN's loss landscape: such failure modes occur when the model becomes trapped in a local optimum located within an extremely sharply descending loss basin [2, 20]. The optimizer typically has difficulty climbing out of this loss basin, resulting in a model that eventually converges to a totally wrong solution. Based on this hypothesis, there should be a significant loss barrier [11] between the failure modes and the true solution of the PDE.

However, our empirical results show that there is no such a loss barrier between the failure mode and the true solution that can block the optimizer from surmounting. As shown in Fig. 1 (a)&(b), we find that the model that eventually converges to an ideal numerical solution also has experienced a failure mode pattern during the training process. There is a similar non-synchronous decrease in residual loss and error during the training process in both success and failure mode cases. As in Fig. 1 (c)&(d), the expected sudden rise in loss in line with the traditional loss landscape understanding does not occur during the transition of the model from a failure mode pattern to a success case.

39th Conference on Neural Information Processing Systems (NeurIPS 2025).

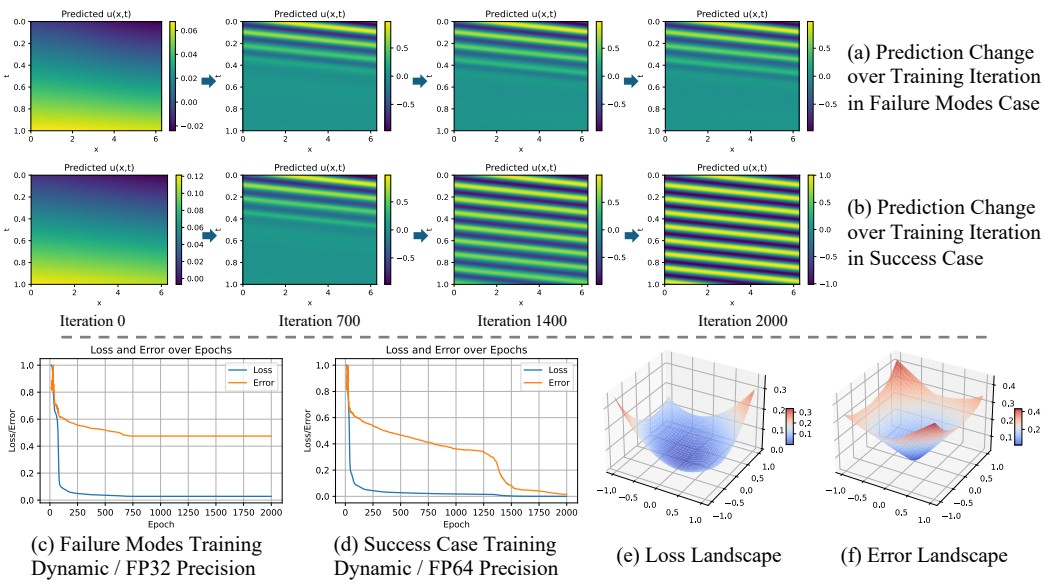

Figure 1: Training Dynamic of PINN's failure mode case and success case.

This phenomenon directly challenges the current understanding of the loss landscape of PINN failure cases in the machine learning community. The cause of the PINN failure case is not a local optimal solution. Instead, as shown in Fig. 1 (e)&(f), there is a significant discrepancy between the loss landscape and error landscape of a PINN. It suggests that the so-called failure mode may just be an intermediate stage of PINN optimization. The model ends up in failure mode due to an unanticipated stop in the optimization process. Therefore, this leads to a fundamental question:

*What makes PINN's optimization process unexpectedly stop at a failure mode?*

We found the answer to this question surprisingly simple: The **arithmetic precision** of the model is not enough to maintain the optimization process any longer. Arithmetic precision is generally considered not to play a decisive role in neural networks in vision and language, where using FP32, FP16 and even lower precision is common practice. But the problem PINN is dealing with is scientific computing with high precision requirements. By using the default precision setting provided in the deep learning framework, the convergence condition of L-BFGS [21], which is the mostly used robust optimizer for PINN, is triggered prematurely due to the lack of enough arithmetic precision. To illustrate that, we show in Fig. 1, with exactly the same problem setting, models with FP32 precision consistently present failure modes, while the models with FP64 precision always succeed.

Regarding the training dynamic, we identify that the training process of PINN will undergo three phases: un-converged phase, failure phase, and success phase. With the same initialization and optimization setup, models with different precisions will finally stop in different stages of optimization. Failure modes problems that used to be considered more difficult (e.g., with a higher-frequency convection parameter) have longer failure phases and more gentle loss plain. As a result, the required arithmetic precision is positively correlated with the difficulty of solving a failure mode problem.

Further, we find that all these known failure modes can be solved by vanilla PINN with sufficient arithmetic precision using the L-BFGS optimizer. Performance of vanilla PINNs with FP64 precision on PDEs like convection can surpass most state-of-the-art model architectures that claim an enhancement on PINN failure modes. With reduced arithmetic precision, models like PINNsFormer [36] show a typical failure mode feature again, indicating that it may not solve the failure modes fundamentally.

**Contributions.** In this paper, we mainly made the following contributions:

- We revolutionize the understanding of PINN failure modes. We reveal that, on the loss landscape of PINN, instead of being in a separate loss basin, the failure mode has a pathway to the optimal solution that can be found by the optimizer.

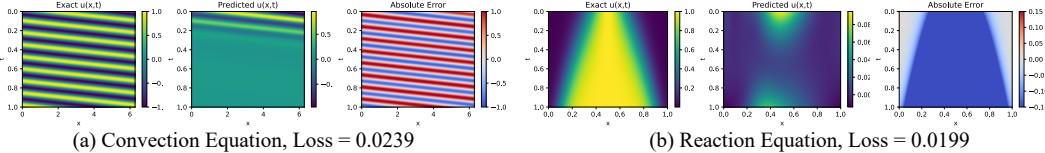

(a) Convection Equation, Loss = 0.0239     (b) Reaction Equation, Loss = 0.0199

Figure 2: Some cases of PINN failure modes.

- We reveal that unexpected stopping of the optimizer due to insufficient arithmetic precision is the core cause of PINN failure modes. With suitable precision, vanilla PINN is strong enough to avoid failure modes with the L-BFGS optimizer.
- We show that the optimization process of PINN is divided into three stages related to difficulty, and different precisions will make the optimization stop at different stages.

## 2 Related Works

**Physics-Informed Neural Networks.** Physics-Informed Neural Networks[27] embed the governing differential equations of a physical system directly into a neural-network training objective. Instead of fitting a network solely to observational data, PINNs minimize a residual loss that penalizes violations of the PDE (or ODE/integro-differential law) at a set of collocation points. This simple idea turns a neural network $u_\theta(x, t)$ into a mesh-free solver that learns a function satisfying

$$\mathcal{F}(u_\theta(x,t)) = 0, \forall (x,t) \in \Omega; \ \mathcal{B}(u_\theta(x,t)) = 0, \forall (x,t) \in \partial\Omega, \tag{1}$$

where $(x,t) \in \Omega$ is the spatial-temporal coordinate, $\partial\Omega$ is the boundary of $\Omega$. The $\mathcal{F}$, and $\mathcal{B}$ denote the operators defined by PDE equations, initial conditions, and boundary conditions, respectively.

The training of PINNs leverages automatic differentiation techniques [25] available within contemporary deep learning frameworks, such as PyTorch [26], to perform numerical differentiation and thereby construct the residual loss at selected collection points. Recent studies have demonstrated that second-order optimization methods, notably quasi-Newton methods such as SSBFGS [24], Broyden [5], and L-BFGS [21], can significantly enhance the stability and fitting accuracy of PINN training [18, 15, 33, 20], compared to relying exclusively on first-order optimizers such as Adam [17].

**Failure Modes in Physics-Informed Neural Networks.** Despite the promising performance and broad applicability, PINNs are susceptible to several failure modes that hinder their effectiveness in practice [20]. The failure modes of PINN refer to a phenomenon in which the PDE residual loss of a model is optimized to a very low level (close to 0), but its resulting solution is still far from the expected true solution. As in Fig. 2, 1-d convection and reaction equations are optimized to close-zero level, but their approximations to the equation's true solutions present the over-simplified patterns.

In response to this phenomenon, researchers have come up with a very wide range of conjectures. Early attempts involve curriculum regularization and sequence-to-sequence learning [20], which breaks down complex PINN problems into simple ones and conquers them one at a time. Subsequently, R3 [9] tries to address this issue with resampling on the failure area. Then, several model architectures like PINNsFormer [36], ProPINN [34], PirateNet [30], and PINNMamba [35] are proposed to help the propagation of the correct pattern among the collection points. From an optimization perspective, researchers also propose regional optimization [33] and Adam+L-BFGS [28] to help the convergence.

Yet, these approaches rely on a hypothesis that the failure mode of PINN is a local minimal that completely isolated from the true solution in loss landscape. During the optimization, model either goes towards a failure mode loss basin or a true solution basin. These approaches intend to induce the model towards the true solutions with an inductive bias. While these methods have achieved some results on some part of failure mode cases, their understanding is not intrinsic. We argue that the failure modes are in the same loss basin as the true solution, and are caused by early stopping.

**Arithmetic Precision in Neural Networks.** Arithmetic precision is commonly thought to have a subsidiary impact on the mainstream applications of neural networks [14]. Although early deep-learning systems relied almost exclusively on 32-bit floating-point arithmetic, a large body of empirical work now shows that reducing numerical precision rarely degrades the accuracy of modern models.

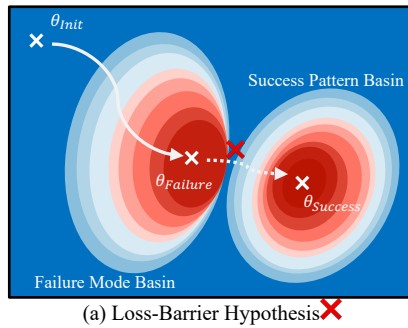 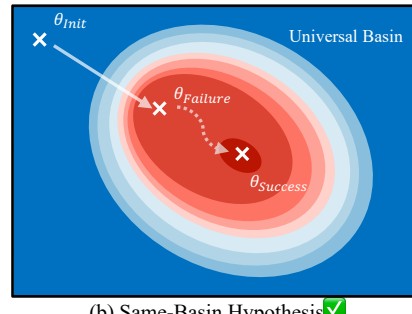

(a) Loss-Barrier Hypothesis ✗          (b) Same-Basin Hypothesis ✅

Figure 3: A schematic that illustrates two kinds of loss landscape hypothesis.

Production-scale vision, speech, and language pipelines routinely deploy INT8/FP8/FP16/BF16 kernels or mixed-precision execution and deliver FP32-level quality with multi-fold improvements in speed and energy efficiency [37, 13, 10]. On the GPU-design side, hardware is now co-engineered for these low-precision formats: NVIDIA's Hopper architecture [8] adds FP8/INT8 Tensor Cores and a Transformer Engine for efficiency-accuracy trade-off. This co-exploration of lower-precision from both software and hardware hints that, in general application of neural networks, arithmetic precision might not be an essential determinant of the model performance. However, we identify that this trend may not fit the scientific machine learning. Numerical methods based on neural networks need to take the same high accuracy calculations as traditional methods (e.g., finite element methods [3]).

## 3 Problem and Experiment Settings

### 3.1 Problem Settings

The PINN employs a neural network parameterized by $\theta$ to approximate the solution of a PDE defined as Eq. 1 by optimizing the following residual loss on $101 \times 101$ collection points $(x_i, t_i)$:

$$\underset{\theta}{\text{minimize}} \quad \mathcal{L}(u_\theta) = \lambda_{\mathcal{F}}\mathcal{L}_{\mathcal{F}}(u_\theta) + \lambda_{\mathcal{B}}\mathcal{L}_{\mathcal{B}}(u_\theta),$$

$$\text{where } \mathcal{L}_{\mathcal{F}}(u_\theta) = \frac{1}{|\chi|} \sum_{(x_i,t_i)\in\chi} \|\mathcal{F}(u_\theta(x_i,t_i)\|^2; \mathcal{L}_{\mathcal{B}}(u_\theta) = \frac{1}{|\partial\chi|} \sum_{(x_i,t_i)\in\partial\chi} \|\mathcal{B}(u_\theta(x_i,t_i)\|^2; \quad (2)$$

To evaluate the performance of the models, we take relative Mean Absolute Error (rMAE, a.k.a $\ell_1$ relative error) and relative Root Mean Square Error (rRMSE, a.k.a $\ell_2$ relative error) formulated as:

$$\text{rMAE}(\hat{u}) = \frac{\sum_{n=1}^{N} |u_\theta(x_n,t_n) - u(x_n,t_n)|}{\sum_{n=1}^{N} |u(x_n,t_n)|}, \ \text{rRMSE}(\hat{u}) = \sqrt{\frac{\sum_{n=1}^{N} |u_\theta(x_n,t_n) - u(x_n,t_n)|^2}{\sum_{n=1}^{N} |u(x_n,t_n)|^2}}, \quad (3)$$

where N is the number of test points, $u(x,t)$ is the ground truth, and $\hat{u}(x,t)$ is the model's prediction.

In this paper, we investigate four famous failure mode PDEs that are widely studied in the PINN community: convection equations, reaction equations, wave equations, and Allen-Cahn [20, 36, 34]. The descriptions and details of these problems can be found in the Appendix A.

### 3.2 Experiments Settings

We initialize the vanilla PINN with an MLP with 3 hidden layers and 512 neurons in each layer. We include advanced PINN neural architectures like PINNsFormer [36], KAN [22], PINNMamba [35] to test the generalization ability of our findings, following their original settings. We train the models with L-BFGS optimizer following common practice [20, 33, 36, 34, 35]. All the experiments are implemented on an NVIDIA H100 GPU, with CUDA version 12.8 and Pytorch vesion 2.1.1.

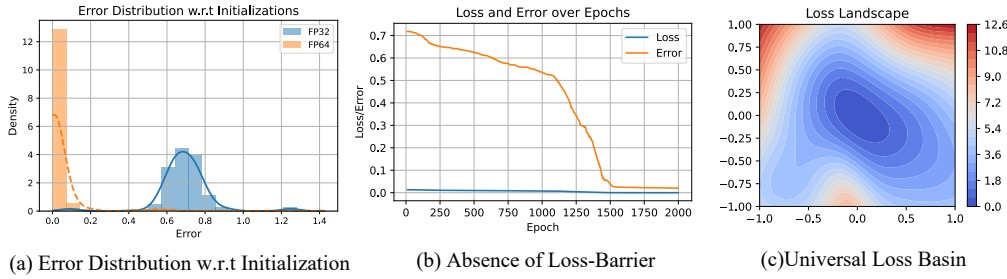

(a) Error Distribution w.r.t Initialization     (b) Absence of Loss-Barrier     (c)Universal Loss Basin

Figure 4: Several empirical results on convection support Same-Basin Hypothesis.

## 4 Understanding PINN Failure Modes from Loss Landscape Perspective

In this section, we demonstrate that the failure modes of PINNs are not attributed to the local minimal with extremely steep loss landscape, which is a view widely accepted by scholars who have studied this area. Instead, we point out that, the failure modes are in the same loss basin as the true solution. The model finally present a failure mode is ascribed to the early stopping of the optimization process.

### 4.1 Loss-Barrier Hypothesis vs Same-Basin Hypothesis: Which is the True Story?

We first state two hypotheses for failure modes of PINNs: (1) the *Loss-Barrier Hypothesis*, which is widely accepted by the PINN researchers, and (2) the *Same-Basin Hypothesis*, which we propose.

**Loss-Barrier Hypothesis.** PINN failures are initially attributed to that "the optimizer has gotten stuck in a local minima with a very high loss function" [20]. This assertion is then been overthrown because the observation of near-zero empirical residual loss in failure modes experiment cases [33, 34, 35, 36]. This hints that the cause of PINN failure modes is not that PDE residual loss cannot be optimized. Instead, as shown in Fig. 3 (a), it seems that the failure modes are because the optimizer is trapped in a local minimum of the loss landscape, which is nearly as low as the true solution's empirical residual loss on collection points. Under this hypothesis, the optimizer is not able to distinguish the failure modes and the true solution minimum. Therefore, models cannot find the true solution, because there is a **loss barrier** between the failure mode local minimal and the true solution minimal. The model's loss faces a huge penalty if its parameters are optimized to cross such a loss barrier.

**Same-Basin Hypothesis.** Based on the intuition that the loss basins are generally connected with a simple perturbation [1], we propose an alternative same-basin hypothesis. It says that the failure modes parameters are in the same loss basin as the true solution, as shown in Fig. 3 (b). There is a large flat plain at the bottom of this loss basin, and the loss of the true solution is only slightly lower than that of the failure modes. Due to some reasons, the optimizer early stops when it reaches the flat plain in the iterated optimization process. In this hypothesis, there should not be a significant loss barrier between the failure mode and the true solution. As a result, the optimizer with a global convergence guarantee should be able to find the global minimum empirically with ideal computational conditions.

### 4.2 Physics-Informed Neural Networks Follow Same-Basin Hypothesis

Although the Loss-Barrier Hypothesis is generally believed to hold, we identify three important clues pointing to the Same-Basin Hypothesis: (1) initialization insensitivity, (2) absence of loss barrier, and (3) loss connectivity. This hints that the failure modes are caused by insufficient optimization.

**PINNs Fail with any Random Initializations.** If the failure mode patterns and the true solution patterns are in separate loss basins, it should be the case that, with some initialization, the model can be optimized to the true solution. But in practice, the situation is that the model is always trapped in the failure mode pattern, no matter how we conduct the random initializations. As shown in Fig.4 (a), with 2000 different random initializations, the trained PINNs always fail on convection equations. This implies the extreme sparsity of the global optimum over the loss landscape. Under this implication, Loss-Barrier Hypothesis completely contradicts the intrinsic lottery ticket phenomena of neural networks [11]. This suggests choosing the alternative hypothesis: Same-Basin Hypothesis.

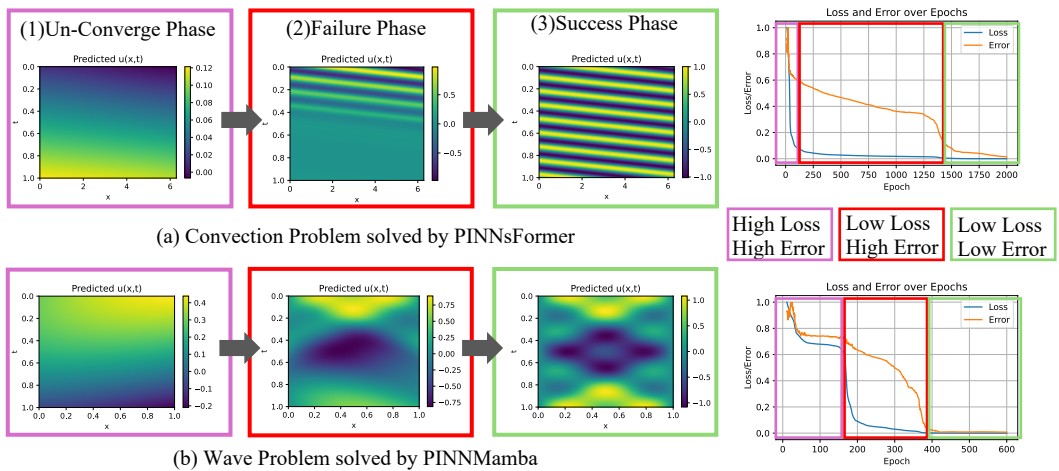

Figure 5: Loss-Error Dynamics of PINN models can be in three phases.

**Non-existent Loss Barrier from Failure Mode to Optimized Solution.** A gravely important finding is that the expected loss barrier in the Loss-Barrier Hypothesis doesn't empirically exist. To illustrate this finding, we initialize the PINN model with failure mode parameters that have been fully optimized for the convection problem, where there is a convergence signal of L-BFGS optimizer. Then we turn the arithmetic precision from FP32 to FP64 (an important finding we will discuss in Section 5), and continue the training process for 2000 iterations. We find that, unlike the loss function that is no longer updated at all as shown in Fig. 1 (c), the loss shows a further trace decline. As shown in Fig. 4 (b), during this further decline process, we observe a steep decline of the error (rMAE). This is direct evidence against the Loss-Barrier Hypothesis. The absence of a loss barrier in practice hints that the failure mode minimal and the true solution minimal are connected in the loss landscape.

**Universal Trough in Loss Landscape.** Further, if the Loss-Barrier Hypothesis holds, there should be a very large number of loss basins that are close to zero at their lowest point. However, as shown in Fig. 4 (c), we can only observe one trough in the loss landscape. The minimum basins are not separated, instead, there is a connected trough in the loss landscape. This directly disproves the assertion that PINN's loss is too rugged, making model optimization difficult. This phenomenon suggests that the failure modes from PINN and the true solution should be in the same basin, and there should be an optimization pathway between them. We further visualized the loss and error landscape of a well-trained model, as shown in Fig. 1 (e)&(f). We found that the loss is relatively level and smooth, and does not have the expected extremely rugged loss landscape. In contrast, the error landscape is much steeper at the position near the true solution optimal. This means that a small perturbation to a parameter can have a much larger effect on the error of the model than the loss. At the same time, the true solution exists in only a very small region on this large loss plain.

The three clues above show, directly or indirectly, that the real reason why PINNs can have failure modes is not an insurmountable barrier in the loss landscape, in other words, it is not due to being stuck in a local optimum. Instead, the PINN true solution of the equation and the failure modes are in the same loss basin. This means that when considering the optimization problem of PINN, the researcher should accept the Same-Basin Hypothesis for model design and tuning.

### 4.3 The Loss-Error Dynamics of PINN's Training

Based on Same-Basin Hypothesis, the whole training dynamics of the PINN model can be divided into three phases: (1) Un-Converged phase, where both the loss and error are still in high level, suggesting the model parameters are still not optimized; (2) Failure phase, where the loss is optimized to near-zero level but the error is still in the high level, and the model presents failure modes such as naive solution; and (3) Success phase, where the model parameter can both achieve near-zero level loss and error. This implies that every model that is optimizable to arrive at a near-true solution also experiences a pattern of failure modes at some point during the training process, yet the model is able

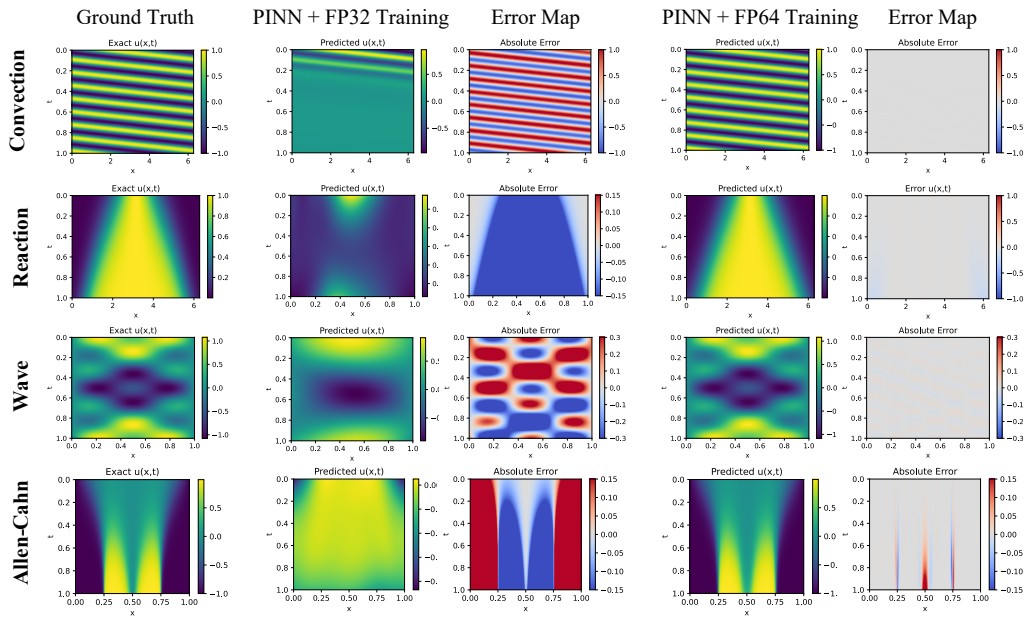

Figure 6: Results for solving 4 failure mode PDE problems with different arithmetic precision.

to be further optimized and eventually arrives at the optimal solution of the equations that the model is capable of giving. Thus, the failure modes are intermediate states of the PINNs training dynamics.

We empirically confirm these 3 phases with different PINN models and PDE problem settings. As shown in Fig. 5, we train a PINNsFormer [36] and a PINNMamba [35], which have been empirically proven to solve the failure modes problem, for the convection and wave problem, respectively. We note that the division of training dynamics of PINN into three phases is common in various problem settings. Different problem settings have different phase duration characteristics. Those PDEs with longer failure phase are more sensitive and likely to be affected by failure modes. When the PINN failure modes were first identified, it was concluded that the difficulty of modeling the PDE increases with the PDE parameter [20]. We attribute this to that the larger PDE parameter will extend the duration of the failure phase. Therefore, the optimization processes are more likely to stop at the failure phase.

## 5 Precision Serves a Key Factor in PINN's Training

We now know that the cause of failure modes is the premature stopping of the model optimization process. This leads to another question: what contributes to the premature stopping of PINN model optimization? In this section, we reveal that the arithmetic precision is a key issue of the PINN optimization. Unlike neural networks in CV and NLP, single-precision (FP32) training does not guarantee that all neural networks corresponding to PDE problems are efficiently optimized. Instead, double-precision (FP64) training is a stable strategy for various model and problem settings.

### 5.1 Failure Modes Disappear when Using Double-Precision Training

We first show that the most basic vanilla PINN can solve all currently known failure modes problems when using double precision training. As shown in Fig. 6, with all these PDE problems where the vanilla PINN exhibits failure modes, the use of double-precision training yields solutions that are highly consistent with ground truth. This suggests that using double-precision training allows the model optimization to go through all three phases, rather than stopping at the failure phase. Moreover, as shown in Table 1, the vanilla PINN trained with double precision can outperform all state-of-the-art methods in terms of prediction error. It hints that the inductive-biased approach that was once based on the understanding of the Loss-Barrier Hypothesis is not fundamental to the addressing of PINN

Table 1: Comparison with baseline methods on 4 failure mode problems.

| Model | Convection | | | Reaction | | |
|---|---|---|---|---|---|---|
| | Loss | rMAE | rRMSE | Loss | rMAE | rRMSE |
| PINN [27] | 0.0133 ± 0.0055 | 0.6904 ± 0.0826 | 0.7640 ± 0.0694 | 0.1991 ± 0.0001 | 0.9788 ± 0.0019 | 0.9778 ± 0.0018 |
| QRes [6] | 0.0153 ± 0.0027 | 0.7498 ± 0.0464 | 0.8184 ± 0.0382 | 0.1991 ± 0.0001 | 0.9826 ± 0.0023 | 0.9830 ± 0.0026 |
| PINNsFormer [36] | 0.0009 ± 0.0001 | 0.0327 ± 0.0068 | 0.0435± 0.0073 | 3.0e-6 ± 1.0e-6 | 0.0147 ± 0.0013 | 0.0296 ± 0.0027 |
| KAN [22] | 0.0250 ± 0.0042 | 0.6213 ± 0.0675 | 0.6985 ± 0.0701 | 7.0e-6 ± 1.0e-6 | 0.0167 ± 0.0014 | 0.0312 ± 0.0034 |
| PirateNet [30] | 0.0347 ± 0.0061 | 0.9704 ± 0.1826 | 0.9740 ± 0.1894 | 4.0e-6 ± 1.0e-6 | 0.0178 ± 0.0023 | 0.0443 ± 0.0064 |
| RoPINN [33] | 0.0189 ± 0.0062 | 0.6251 ± 0.0940 | 0.7204 ± 0.0941 | 4.8e-5 ± 9.0e-6 | 0.0589 ± 0.0161 | 0.0965 ± 0.0310 |
| PINNMamba [35] | 0.0001 ± 2.0e-5 | 0.0184 ± 0.0037 | 0.0197 ± 0.0038 | **1.0e-6 ± 1.0e-6** | **0.0092 ± 0.0017** | **0.0213 ± 0.0036** |
| PINN_FP64 | **5.0e-6 ± 1.0e-6** | **0.0059 ± 0.0013** | **0.0072 ± 0.0017** | 1.0e-5 ± 5.0e-6 | 0.0271 ± 0.0063 | 0.0502 ± 0.0111 |

| Model | Wave | | | Allen-Cahn | | |
|---|---|---|---|---|---|---|
| | Loss | rMAE | rRMSE | Loss | rMAE | rRMSE |
| PINN [27] | 0.0174 ± 0.0061 | 0.2746 ± 0.0574 | 0.2837 ± 0.0571 | 0.4703 ± 0.2986 | 0.9720 ± 0.0370 | 0.9662± 0.0300 |
| QRes [6] | 0.0965 ± 0.0192 | 0.5335 ± 0.1230 | 0.5273 ± 0.1172 | 0.9887 ± 0.0021 | 0.9821 ± 0.0089 | 0.9846 ± 0.0092 |
| PINNsFormer [36] | 0.0231 ± 0.0017 | 0.3492 ± 0.0871 | 0.3571 ± 0.0872 | 0.4625 ± 0.2875 | 0.9908 ± 0.0401 | 0.9913 ± 0.0420 |
| KAN [22] | 0.0067 ± 0.0012 | 0.1475 ± 0.0354 | 0.1489 ± 0.0357 | 0.0234 ± 0.0031 | 0.3166 ± 0.0233 | 0.5661 ± 0.0440 |
| PirateNet [30] | 0.0153 ± 0.0051 | 0.2544 ± 0.0471 | 0.2637 ± 0.0480 | 0.0017 ± 0.0001 | 0.1088 ± 0.0109 | 0.1889 ± 0.0180 |
| RoPINN [33] | 0.0015 ± 0.0005 | 0.0631 ± 0.0226 | 0.0642 ± 0.0238 | - | - | - |
| PINNMamba [35] | 0.0002 ± 3e-5 | 0.0193 ± 0.0033 | 0.0195 ± 0.0033 | 0.0027 ± 0.0002 | 0.1432 ± 0.0123 | 0.2645 ± 0.0201 |
| PINN_FP64 | **4.2e-5 ± 1.6e-5** | **0.0080 ± 0.0032** | **0.0081 ± 0.0031** | **1.3e-5±4.0e-6** | **0.0157 ± 0.0036** | **0.0545 ± 0.0112** |

failure modes. Instead, insufficient arithmetic precision is a straightforward contributor to the failure modes of the PINN. As with traditional numerical methods such as Finite Element Analysis, PINN also requires the use of double precision to achieve accurate and trustworthy results.

## 5.2 Optimization Stops Early without Sufficient Arithmetic Precision

Based on the analysis in Section 4, the failure modes of PINN are due to premature stopping of the optimization process. In order to find the reason why arithmetic precision affects the optimization stopping of the PINN, we target the optimizer and its implementation in a deep learning framework.

**PINNs Commonly Use L-BFGS Optimizers.** The PDE residual contains high-order differential operators whose spectra can span many orders of magnitude, giving a Hessian with extreme eigenvalue ratios. First-order methods (Adam/SGD) therefore move in directions dominated by the largest eigenvalues and make little progress on the stiff directions. Rightly, second-order optimizers such as Newton, BFGS, and L-BFGS implicitly (or explicitly) invert an estimate of the Hessian, scaling each parameter update by the local curvature and thereby shrinking the stiff directions while enlarging the flat ones. Among all the second-order optimizers, L-BFGS is the most commonly utilized for its excellent performance while maintaining relatively low-level time and space complexity. L-BFGS [21] approximates the inverse Hessian using a history of the past m updates of the position and gradient. At each iteration, it constructs the Hessian approximation implicitly using the stored history. We include the details of the L-BFGS optimizer in Appendix B.

**Existing Implementation of L-BFGS Optimizer.** In deep learning framework's implementation of the L-BFGS algorithm, such as 'torch.optim.LBFGS', algorithmic control is divided between user-defined outer loops and library-defined inner loops. The user governs the outer loop by repeatedly invoking 'optimizer.step(closure)', where each call corresponds to a single quasi-Newton update that assimilates one curvature pair for the inverse Hessian. During a single call to 'optimizer.step(closure)', it may perform up to 'max_iter' inner loops that repeatedly reevaluate the objective and gradient, compute a quasi-Newton search direction via the two-loop recursion with the latest 'history_size' curvature pairs, optionally conduct a Strong-Wolfe line search [31], and update both parameters and curvature history. The inner loop halts when convergence criteria on gradient norm or successive parameter changes are met, after which control returns to the user-level outer loop.

**Low-Precision Causes the Inner Loop Early Stopping.** We found that insufficient arithmetic precision prematurely triggers the convergence criteria of the inner loop. As shown in Fig. 7 (a), FP32 triggers the inner loop early stopping (the count of inner loops <= 5) much earlier than FP64 does. This is due to the fact that the trigger condition for convergence in PINN ('tolerance_change') has a value of 1e-7 that is smaller than the machine unit $\varepsilon$ for single precision floating point numbers. Machine unit $\varepsilon$ is the smallest positive floating-point number that, when added to 1, yields a representable

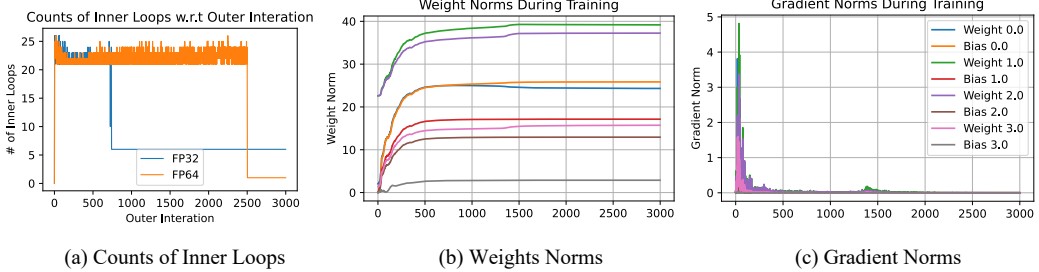

| | | | | | |
|---|---|---|---|---|---|
| | (a) Counts of Inner Loops | | (b) Weights Norms | | (c) Gradient Norms |

Figure 7: The values of the PINN weights are orders of magnitude different than the values of the gradients, and the ratio is also incremental, causing single precision to underflow earlier.

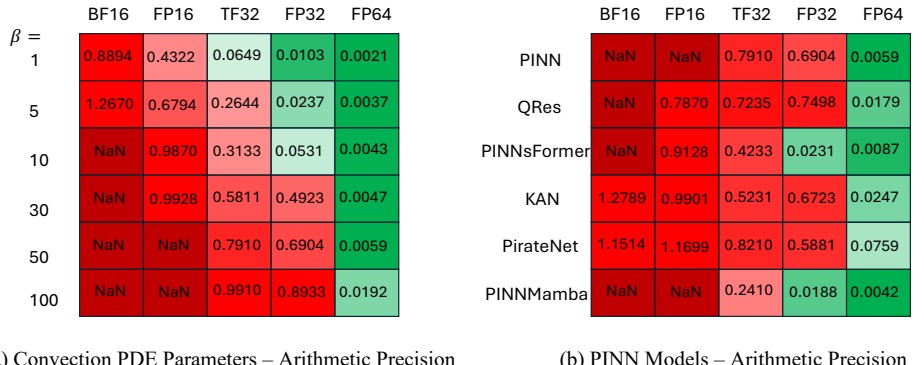

(a) Convection PDE Parameters – Arithmetic Precision  (b) PINN Models – Arithmetic Precision

Figure 8: Results on convection equations with different arithmetic precision: (a) PINN's rMAE on various convection parameters $\beta$. (b) Different models' rMAE on convection with $\beta = 50$.

value strictly greater than 1 on a given hardware/format:

$$\varepsilon \;=\; \min\{\,\varepsilon > 0 : 1 + \varepsilon \text{ is representable and } 1 + \varepsilon \neq 1\}. \tag{4}$$

For a radix $\beta$ (= 2) and $p$ binary digits in the significand, $\varepsilon = \frac{1}{2}\beta^{1-p}$, which means the FP32 has the machine $\varepsilon = 1.19$ e-7 > 'tolerance_change' = 1e-7, while FP64's machine $\varepsilon = 2.22$ e-16.

As shown in Fig. 7 (b), during the optimization of PINN, the Norms of the weights are incrementally increasing in the order of magnitude of 1e+1 level, which is greater than 1. This means that the computation effective numerical precision is even more coarse-grained than machine $\varepsilon$, and thus computing the changes of weights will be easier to underflow. In Fig. 7 (c), we show that the norms of the gradient are decreasing to near-zero level gradually, which will cause the PINN optimization problem to be extremely ill-conditioned. Therefore, the amount of weight change can quickly drop below machine $\varepsilon$ with FP32 training, since the weight change is positively correlated with gradient.

### 5.3 Harder Equations Require Higher Arithmetic Precision

It has been shown that, for a parameterized PDE, the PINN optimization of the equations becomes progressively more difficult as the parameters of the equations become larger, and even failure modes occur [20, 7]. The reason for this phenomenon is that the frequency of model changes will become higher, and the pattern of the model will become more versatile. We attribute this to the fact that as the parameter of the equation gets larger, the separation of its error and loss landscapes also gets progressively larger. This means that for more difficult problems with a larger parameter, an early stopping point in the optimization process will result in a larger error. As shown in Fig. 8 (a), we confirm this problem's complex trend with varying degrees of arithmetic precision from BF16 to FP64. We find that for more complex equation problems (like $\beta = 100$), increasing the arithmetic precision of the model to double precision can effectively solve the failure modes of PINN.

Table 2: Comparison of FP32 and FP64 configurations in terms of relative RMSE, training time per iteration, and memory usage across PDE benchmarks.

| FP32 | | | |
|---|---|---|---|
| Equation | rRMSE | Training Time/Iter (s) | Memory (MB) |
| convection | 0.7640 | 0.29 | 1609 |
| reaction | 0.9778 | 0.26 | 1629 |
| wave | 0.2837 | 0.47 | 2295 |
| Allen-Cahn | 0.9662 | 0.40 | 1975 |
| FP64 | | | |
| Equation | rRMSE | Training Time/Iter (s) | Memory (MB) |
| convection | 0.0072 | 0.32 | 2441 |
| reaction | 0.0502 | 0.29 | 2481 |
| wave | 0.0081 | 0.61 | 3845 |
| Allen-Cahn | 0.0545 | 0.48 | 3167 |

## 5.4 Arithmetic Precision's Impact on Various Model Architectures.

In order to verify whether the multiple understandings of model architectures [36, 35] claiming to address PINN failure modes are intrinsic, and whether our proposed method for improving numerical precision is orthogonal to other methods, as shown in Fig. 8, we evaluate a lot of models on convection equations with parameter $\beta$ being set to 50. We find that just a slight decrease in the precision of PINNsFormer and PINNMamba (from FP32 to TF32) can make the failure modes reproduce, which suggests that the method of constructing inductive bias based on the Loss-Barrier Hypothesis is not an intrinsic solution to the failure modes problem. Using double-precision training, all models converge to the correct solution, which illustrates that our understanding of loss landscape and model precision holds the correctness, generalizability, and orthogonality to other methods.

## 5.5 Discussion of Computation

As shown in Table 2, we evaluate the FP32 and FP64 training overhead on a Nvidia H100 PCIe GPU. The FP64 latency per iteration is roughly 1.1-1.3 times that of FP32, and the GPU memory footprint is about 1.5-1.7 times that of FP32. This is due to the fact that, with Tensor Core, the FP64 peak arithmetic on H100 PCIe is 60 TFLOPS, the same as the FP32 peak arithmetic. As a result, FP64 is not very much slower than FP32. We observe about 10-30% more latency of FP64 than FP32 that we believe is due to larger memory accesses. This indicates that training with FP64 does incur an increase in computational overhead, but under modern GPU architectures the increase is marginal.

## 6 Conclusion

In this paper, through hypothesis testing on the loss landscape of PINN, we demystify that the real cause of PINN's failure modes is the early stopping of the optimization process. We reveal that this early stopping phenomenon can be easily solved by improving the training arithmetic precision of the model. Training with double precision will enable the vanilla PINN to beat state-of-the-art PINN model architectures on various PDE problem settings. This prompts the broader AI for Science community that PINN is still essentially a numerical solution method for PDEs, and that computational precision plays just as crucial a role as it does in traditional methods.

## Acknowledgements

This work is supported, in part, by the National Science Foundation and the Institute of Education Sciences under Grant 2229873 (AI4ExceptionalEd), National Science Foundation under Grant 2235364 (FuSe-TG), and SUNY-IBM AI Collaborative Research Award. Any opinions, findings and conclusions or recommendations expressed in this material are those of the author(s) and do not necessarily reflect the views of the sponsors.

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

# A PDE Setups

## A.1 1-D Convection

The one–dimensional convection (or advection) equation characterises the transport of a scalar field $u(x, t)$—such as temperature, concentration, or momentum—under a uniform velocity $\beta$. Widely studied in fluid dynamics and transport theory, it is expressed as

$$\frac{\partial u}{\partial t} + \beta \frac{\partial u}{\partial x} = 0, \ \forall x \in [0, 2\pi], t \in [0, 1],$$
$$u(x, 0) = \sin x, \ \forall x \in [0, 2\pi], \tag{5}$$
$$u(0, t) = u(2\pi, t), \ \forall t \in [0, 1],$$

where $\beta$ denotes the constant advection speed. Larger values of $\beta$ increase the difficulty for PINNs, making this equation a common benchmark with known failure modes. Following prevailing practice [36, 33], we set $\beta = 50$.

Its analytical solution is

$$u_{\text{ana}}(x, t) = \sin(x - \beta t). \tag{6}$$

## A.2 1-D Reaction

The one–dimensional reaction equation models the temporal evolution of a reacting species along a single spatial dimension:

$$\frac{\partial u}{\partial t} - \rho u(1 - u) = 0, \ \forall x \in [0, 2\pi], t \in [0, 1],$$
$$u(x, 0) = \exp\left(-\frac{(x - \pi)^2}{2(\pi/4)^2}\right), \ \forall x \in [0, 2\pi], \tag{7}$$
$$u(0, t) = u(2\pi, t), \ \forall t \in [0, 1],$$

with $\rho$ representing the growth-rate coefficient. Increasing $\rho$ likewise poses greater challenges for PINNs; we adopt the standard choice $\rho = 5$ in accordance with [36, 33]. The closed-form solution is

$$u_{\text{ana}} = \frac{\exp\left(-\frac{(x-\pi)^2}{2(\pi/4)^2}\right) \exp(\rho t)}{\exp\left(-\frac{(x-\pi)^2}{2(\pi/4)^2}\right)\left(\exp(\rho t) - 1\right) + 1}. \tag{8}$$

## A.3 1-D Wave

The one–dimensional wave equation describes wave propagation—for example, vibrations on a string—and in our study takes the form

$$\frac{\partial^2 u}{\partial t^2} - 4\frac{\partial^2 u}{\partial x^2} = 0, \ \forall x \in [0, 1], t \in [0, 1],$$
$$u(x, 0) = \sin(\pi x) + \frac{1}{2}\sin(\beta \pi x), \ \forall x \in [0, 1], \tag{9}$$
$$\frac{\partial u(x, 0)}{\partial t} = 0, \ \forall x \in [0, 1],$$
$$u(0, t) = u(1, t) = 0, \ \forall t \in [0, 1],$$

where $\beta$ controls the second harmonic; we set $\beta = 3$ following [36, 33]. Because the PDE involves second-order derivatives and the initial condition contains first-order derivatives, optimisation is notoriously difficult [33]. Nevertheless, this case demonstrates that PINNMamba's matrix-defined time differentiation—with uniform scaling across derivative orders—can better capture temporal dynamics.

Its analytical solution is

$$u_{\text{ana}}(x, t) = \sin(\pi x)\cos(2\pi t) + \sin(\beta \pi x)\cos(2\beta \pi t). \tag{10}$$

## A.4 Allen–Cahn

The Allen–Cahn equation is a canonical reaction–diffusion benchmark:

$$\frac{\partial u}{\partial t} - 0.0001\frac{\partial^2 u}{\partial x^2} + 5u^3 - 5u = 0, \ x \in (-1,1), \ t \in (0,1),$$
$$u(x,0) = x^2\cos(\pi x), \ x \in [-1,1], \tag{11}$$
$$u(-1,t) = u(1,t), \ t \in [0,1],$$
$$\frac{\partial u(-1,t)}{\partial x} = \frac{\partial u(1,t)}{\partial x}, \ t \in [0,1].$$

Because no closed-form analytic solution exists, we adopt a high-resolution spectral approximation [19] as reference data, following [27, 34]. The sharp interface and double entrance makes this PDE a widely used stress test for PINNs and a representative failure mode.

# B  L-BFGS Optimizer

For an unconstrained, twice-differentiable objective $f : \mathbb{R}^n \to \mathbb{R}$, Newton's method uses the exact Hessian $H_k = \nabla^2 f(x_k)$ to obtain the step

$$x_{k+1} = x_k - \alpha_k H_k^{-1} \nabla f(x_k), \tag{12}$$

yielding a quadratic local model and (under standard assumptions) quadratic convergence. Quasi-Newton methods avoid forming or inverting the expensive $H_k$ by maintaining an approximation $B_k \approx H_k$ (or its inverse $H_k \approx B_k^{-1}$) that is updated from first-order information only. The most successful member is the BFGS update

$$H_{k+1} = (I - \rho_k s_k y_k^\top) H_k (I - \rho_k y_k s_k^\top) + \rho_k s_k s_k^\top, \quad \rho_k = \frac{1}{y_k^\top s_k}, \tag{13}$$

with $s_k = x_{k+1} - x_k$ and $y_k = \nabla f(x_{k+1}) - \nabla f(x_k)$. BFGS enjoys global convergence and super-linear local convergence, but it stores a dense $n \times n$ matrix—requiring $O(n^2)$ memory and time per iteration—making it impractical when $n$ is large.

L-BFGS eliminates the quadratic storage by discarding all but the most recent $m$ curvature pairs $\{(s_i, y_i)\}_{i=k-m}^{k-1}$. Instead of storing $H_k$ explicitly, it represents the inverse Hessian implicitly through these vectors and a diagonal scaling $H_k^0$ (often $\gamma_k I$ with $\gamma_k = \frac{s_{k-1}^\top y_{k-1}}{y_{k-1}^\top y_{k-1}}$).

In compact format, let:

$$S_k = [s_{k-m}, \ldots, s_{k-1}] \in \mathbb{R}^{n \times m}, \quad Y_k = [y_{k-m}, \ldots, y_{k-1}], \quad R_k = S_k^\top Y_k \ \text{(upper-triangular)}, \tag{14}$$

$$D_k = \operatorname{diag}(s_{k-m}^\top y_{k-m}, \ldots, s_{k-1}^\top y_{k-1}). \tag{15}$$

With an initial diagonal scaling $H_k^0 = \gamma_k I$ (usually $\gamma_k = \frac{s_{k-1}^\top y_{k-1}}{y_{k-1}^\top y_{k-1}}$), the inverse Hessian can be written without loss of information as

$$H_k = H_k^0 + \begin{bmatrix} S_k & H_k^0 Y_k \end{bmatrix} \begin{bmatrix} R_k^{-\top}(D_k + Y_k^\top H_k^0 Y_k)R_k^{-1} & -R_k^{-\top} \\ -R_k^{-1} & 0 \end{bmatrix} \begin{bmatrix} S_k^\top \\ Y_k^\top H_k^0 \end{bmatrix}. \tag{16}$$

The memory requirement reduces to $O(nm)$ floats, while the computational cost per iteration becomes $O(nm)$ flops. This scaling is linear in the number of parameters when $m$ is fixed, with typical values $m \in [5, 20]$.

## C   Broader Social Impact

Making FP64 the default for PINNs turns a previously brittle solver into a dependable scientific tool: the same vanilla architecture that failed at FP32 converges robustly on convection, reaction, wave and Allen-Cahn equations when upgraded to FP64. Reliable neural PDE solvers promise faster, mesh-free simulations for climate modelling, renewable-energy design, and personalised biomedical devices, potentially shortening innovation cycles and widening access to high-fidelity analysis in domains where traditional finite-element codes are prohibitively slow.

The work also reframes precision as a first-class hyper-parameter in numerical machine learning, aligning PINNs with classical finite-element practice where double precision is standard for safety-critical engineering. This alignment reduces the risk of deploying under-resolved models in high-stakes settings. Yet the same fidelity could be misapplied—for instance, to accelerate the design of destructive fluid-dynamic systems. We therefore release code under a license prohibiting military applications and encourage transparent reporting of energy usage and precision settings to foster reproducibility and responsible adoption.

## D   Limitations

However, double precision is more energy-intensive and currently restricted to high-end hardware such as NVIDIA's H100 GPUs, on which our experiments were run. Increased demand for FP64 compute could raise the carbon footprint of scientific machine learning and exacerbate resource inequity between well-funded labs and those with limited budgets. Conversely, our findings may motivate hardware–software co-design efforts that deliver energy-efficient FP64 pipelines, ultimately mitigating this environmental cost.

