# OpenReview forum: "FP64 is All You Need: Rethinking Failure Modes in Physics-Informed Neural Networks"
_NeurIPS.cc/2025/Conference — NeurIPS 2025 poster_

### Official Review · Reviewer_wrna · 2025-06-16

**Clarity:** 3
**Significance:** 2
**Originality:** 3
**Rating:** 4
**Confidence:** 5

**Summary:**

This paper restudied the cause of PINN's training by L-BFGS optimizer is the early stopping of the optimization process.
 The low-precision such as FP32 causes the inner loop of L-BFGS optimizer early stopping.
On the other hand, by simple replacing FP 32 by double precision FP64 can surprisingly make vanilla PINN convergent for various PDE problems.

**Questions:**

1. Is the training failure caused by the L-BFGS optimizer? Does the same training failure exist for Adam optimizer of PINN?

2. How does the training challenge for the long time prediction of PINN? This work only predict to time $t=1$, how about the long time prediction such as $t=10$, $t=50$?

3. Although the four equations (convection equations, reaction equations, wave equations, and Allen-Cahn) have been widely studied in the PINN community, I don't think they are suitable for bechmarks nowdays. These four equations can be well-solved by PINN's variants, as well as traditional numerical methods. There exists PINN variants and training methods [1] which are superior than the baseline models compared in this paper. Complex PDE problems such as the Navier-Stokes flow with vortex, high dimensional PDEs, inverse PDE problems, etc, should be compared to show the improvements by using high-precision.

[1]Wang S, Sankaran S, Wang H, et al. An expert's guide to training physics-informed neural networks[J]. arXiv preprint arXiv:2308.08468, 2023.

**Ethical Concerns:**

["NO or VERY MINOR ethics concerns only"]

**Final Justification:**

Most of my concerns have been resolved. The method is simple and effective, deserves to be seen in the PINNs' community.

However, I think the widely used Adam optimizer deserves more exploration. The paper still need improvement on experiments.

So I change the rating from 3 to 4, without higher scores.

**Limitations:**

The limitations are stated in Appendix D.

**Quality:**

2

**Strengths And Weaknesses:**

The strengths of this paper is obvious, it improves the PINN training by L-BFGS optimizers simply and significantly.
The phenomenon is studied and analyzed adequately.
However, my concern is whether double-precision changes the training failures by first order optimizers such as Adam.
The authors only concerned the L-BFGS optimizers on commonly studied PDE problems which have been well solved by PINN's variants.
These problems can also be easily solved by traditional numerical methods, and it can not reflect the advantages of PINNs.

---

> ### Author Rebuttal · Authors · 2025-07-31
>
> We sincerely appreciate the constructive comments from reviewer wrna and the time spent on reviewing this paper. We address the questions and clarify the issues accordingly as described below.
>
> >**[Weakness 1]**: However, my concern is whether double-precision changes the training failures by first order optimizers such as Adam.
>
> **[Response to W1]**: The PINN optimized by Adam using double precision has smaller loss, rMAE, and rRMSE than the single-precision case. This indicates that double precision is also useful for first-order optimizers. We show the results on convection equations optimized with Adam for 100000 iteration in the following table.
>
> |Optimizer|Precision|Training Loss|rMAE|rRMSE|
> |---|---|---|---|---|
> |Adam|FP32|0.095331|0.9022|0.9256|
> ||FP64|0.000174|0.1363|0.1716|
> |L-BFGS|FP32|0.0133|0.6904|0.7640|
> ||FP64|5e-6|0.0059|0.0072|
>
> Despite having lower loss and error, we find that the Adam-optimized model still fails to fully address failure modes with double-precision training.
> Several studies have demonstrated the superiority of second-order optimizers over first-order optimizers for PINN. [1,2] We strongly recommend using L-BFGS or Natural Gradient Descent for PINN optimization.
>
>
>
> >**[Weakness 2]**: The authors only concerned the L-BFGS optimizers on commonly studied PDE problems which have been well solved by PINN's variants.
>
> **[Response to W2]**: The resolving of PINN's variant for PINN failure modes is not intrinsic, as shown in Fig. 8(b), these PINN's variant still exhibit failure modes with reduced computational precision. The success of these variants is due to the introduction of some sort of prior, or inductive bias, into the model architecture of these variants. These prior and inductive bias are not necessarily generalizable. As shown in Table 1, these variants usually solve only part of the failure mode problems.
>
> The contribution of our paper is not only to provide a solution to failure modes, but also, we provide an understanding of PINN failure modes. We show how failure modes of PINN should be properly understood from an optimization point of view. The PDE problems we have selected are all widely recognized as having failure modes.[4,5,6,7,9] PINN failure mode is a special phenomenon, usually with low training loss and high training error, which is different from typical optimization problems (high loss, high error) and generalization problems (low training loss, high inference loss). The PDE problems we study are highly representative of failure modes.
>
>
> >**[Weakness 3]**: These problems can also be easily solved by traditional numerical methods, and it can not reflect the advantages of PINNs.
>
> **[Response to W3]**: Our goal in this paper is not to show the advantages of PINN, but to address the real-world problems encountered by PINN. In terms of the advantages of PINN, PINN uses neural networks as a general function approximator to unify physical conservation and data supervision into a microscopic framework through automatic discretization, and thus is more flexible, mesh-saving, and easy to integrate compared to traditional numerical methods in complex geometries, real-time multiple solutions, and physics-data fusion scenarios. Despite these advantages, PINN still has some failure modes, and this paper aims to understand the fundamental reasons for these failure modes.
>
> >**[Question 1]**: Is the training failure caused by the L-BFGS optimizer? Does the same training failure exist for Adam optimizer of PINN?
>
> **[Response to Q1]**: The failure modes of the PINN are not caused by the L-BFGS optimizer, but the hard-to-optimize loss surface. The same training failure exist for Adam as well. In fact, the L-BFGS optimizer has a generally stronger performance on PINN than first-order optimizers such as Adam. The experiments results we have provided in the table above (response to W1) illustrates this.
>
>
> >**[Question 2]**: How does the training challenge for the long time prediction of PINN? This work only predict to time t = 1, how about the long time prediction such as t = 10 , t = 50 ?
>
> **[Response to Q2]**: PINN is agnostic to the length of time, no matter how long training interval t is. Our choice of training interval t = 1 is due to the normalization to the time coordinate, which can be done by a simple linear transformation for the system, and similarly, an inverse transformation can be performed to fit any time coordinate.
>
> The standard PINN method does not support out-of-domain generalization for the time being, and the standard PINN trained on the interval of time $[0, 1]$ does not support generalization on the interval $[1, +\infty]$. This generalization outside of the training domain has been studied in a few studies [3], but is beyond the scope of this paper, and we leave it for future work.
>
> >**[Question 3]**: Although the four equations (convection equations, reaction equations, wave equations, and Allen-Cahn) have been widely studied in the PINN community, I don't think they are suitable for bechmarks nowdays. These four equations can be well-solved by PINN's variants, as well as traditional numerical methods. There exists PINN variants and training methods which are superior than the baseline models compared in this paper. Complex PDE problems such as the Navier-Stokes flow with vortex, high dimensional PDEs, inverse PDE problems, etc, should be compared to show the improvements by using high-precision.
>
> **[Response to Q3]**: The object of this paper is PINN failure modes, and the four benchmarks we use in this paper are classical failure modes problems. These questions are also widely used in several other recent papers that study PINN failure modes.[4,5,6,7,9]
>
> The core goal of this paper is not to reach a new state-of-the-art on benchmarks, but to present an understanding of the particular phenomenon of PINN failure modes. 2-D Navier-Stokes with vortex, Burger's equation, etc., for example, are not PINN failure modes problems, and standard PINNs can achieve good performance with the L-BFGS optimizer under these problems. Therefore, testing on these problems does not reflect the innovation of this paper.
>
> Still, we agree that it would be helpful to test the effectiveness of FP64 training on some complex problems. We present some experimental results in the table below. We find that FP64 still consistently improves the performance of the model.
>
>
> |Problem|Precision|rMAE|rRMSE|
> |---|---|---|---|
> |2D Naiver-Stoke|FP32|13.08|9.08|
> ||FP64|0.0397|0.0285|
> |Fluid Dynamics|FP32|0.3659|0.4082|
> ||FP64|0.1475|0.1721|
>
> The PDE problem are defined as following:
>
> 2D Naiver-Stoke:
>
>
> $\frac{\partial u}{\partial t}+\left(u \frac{\partial u}{\partial x}+v \frac{\partial u}{\partial y}\right)  =-\frac{\partial p}{\partial x}+0.01\left(\frac{\partial^2 u}{\partial x^2}+\frac{\partial^2 u}{\partial y^2}\right)$
>
> $\frac{\partial v}{\partial t}+\left(u \frac{\partial v}{\partial x}+v \frac{\partial v}{\partial y}\right)  =-\frac{\partial p}{\partial y}+0.01\left(\frac{\partial^2 v}{\partial x^2}+\frac{\partial^2 v}{\partial y^2}\right) $
>
> $
> \frac{\partial u}{\partial x}+\frac{\partial v}{\partial y}  =0$
>
> where $u,v$ denotes the velocity on the x and y-axis respectively, and $p$ denotes the pressure field.
>
> Fluid Dynamics:
>
> $\frac{\partial w}{\partial t}+\left(u \frac{\partial w}{\partial x}+v \frac{\partial w}{\partial y}\right)  =\frac{1}{\operatorname{Re}}\left(\frac{\partial^2 w}{\partial x^2}+\frac{\partial^2 w}{\partial y^2}\right), \quad(t, x, y) \in[0, T] \times \Omega $
>
> $ \frac{\partial u}{\partial x}+\frac{\partial v}{\partial y}  =0, \quad(t, x, y) \in[0, T] \times \Omega $
>
> $ w(0, x, y)  =w_0(x, y),(x, y) \in \Omega$
>
> where $u,v$ denotes the velocity  of the fluid on the x and y-axis respectively, $w$ is the vorticity, $T=10$ and $\Omega = [0,2\pi]^2$. $Re = 100$ is Reynolds number, $w_0$ is initial condition.
>
> In addition, we have some concerns about the generalizability of the methods in the paper [8], and the methods they propose do not essentially solve the failure modes. For example, for the convection problem, a periodic prior is introduced to the solution of the equations in [8], but considering that the analytical solution of this problem can be as simple as $u(x,t) = sin(x-\beta t)$, it is clear that this is not generalizable, as well as that this prior assumption of periodicity is too strong.
>
> [1] Rathore P, Lei W, Frangella Z, et al. Challenges in training pinns: A loss landscape perspective. ICML, 2024.
>
> [2] Urbán J F, Stefanou P, Pons J A. Unveiling the optimization process of physics informed neural networks: How accurate and competitive can PINNs be? Journal of Computational Physics, 2025.
>
> [3] Bonfanti A, Santana R, Ellero M, et al. On the generalization of pinns outside the training domain and the hyperparameters influencing it. Neural Computing and Applications, 2024.
>
> [4] Zhao Z, Ding X, Prakash B A. Pinnsformer: A transformer-based framework for physics-informed neural networks. ICLR, 2024.
>
> [5] Wu H, Luo H, Ma Y, et al. Ropinn: Region optimized physics-informed neural networks. NeurIPS, 2024.
>
> [6] Wu H, Ma Y, Zhou H, et al. Propinn: Demystifying propagation failures in physics-informed neural networks[J]. arXiv preprint arXiv:2502.00803, 2025.
>
> [7] Xu C, Liu D, Hu Y, et al. Sub-sequential physics-informed learning with state space model. ICML, 2025.
>
> [8] Wang S, Sankaran S, Wang H, et al. An expert's guide to training physics-informed neural networks[J]. arXiv preprint arXiv:2308.08468, 2023.
>
> [9] Krishnapriyan A, Gholami A, Zhe S, et al. Characterizing possible failure modes in physics-informed neural networks. NeurIPS, 2021.

---

> > ### Comment · Reviewer_wrna · 2025-08-02
> >
> > Thanks for the detailed responses.
> >
> > 1(W1).Although L-BFGS is usually used in PINNs, it is harder for large neural networks when we use PINNs to solve complex PDEs or the advanced physics-informed neural operator cases. So I still think the widely used first order Adam optimizers deserve more exploration.
> >
> > 2(Q2).The training difficulties for large t cannot be simply by normalization(t=50 results in a scale of $\frac{1}{50}\partial_t u$ to t=1, and the PDE changes). I didn't refer to the out-of-domain generalization problem.
> >
> > Despite this, I appreciate the perspective of this work on PINN's failure modes on optimization precisions. The method is clear ,simple and effective, deserves to be seen in the PINN's community. So I increase my score.

---

> > > ### Author Response · Authors · 2025-08-04
> > >
> > > Thank you for your response. We appreciate your support in the acceptance of our paper. If you have any further concerns and questions, we are willing to discuss them with you.
> > >
> > > For the first-order optimizer like Adam, we are working on the additional experiment and will add a discussion in the final version.
> > >
> > > For the larger t problem, actually the Fluid-Dynamic problem we add in rebuttal stage has a larger t=10. We will also explore more kinds of PDEs.
> > >
> > > Thank you for your advice!
> > >
> > > Authors

---

### Official Review · Reviewer_QHbM · 2025-07-01

**Clarity:** 3
**Significance:** 3
**Originality:** 3
**Rating:** 5
**Confidence:** 4

**Summary:**

This paper explores "failure modes" in physics-informed neural networks, a phenomenon commonly blamed for local optima separated from the true solution by steep loss barriers. The authors instead raises the "same-basin hypothesis" stating that the failure modes parameters are in the same loss basin as the true solution. The authors further analyze why failure modes exist with the commonly used L-BFGS optimizer, and find that the coarse machine precision of FP32 is to blame. The authors empirically demonstrate their hypothesis by replacing the FP32 training with FP64 training, and show that FP64 resolves all model's failure modes in a convection equation case.

**Questions:**

No.

**Ethical Concerns:**

["NO or VERY MINOR ethics concerns only"]

**Limitations:**

No.

**Paper Formatting Concerns:**

No.

**Quality:**

3

**Strengths And Weaknesses:**

Strengths

- Clear motivation: the paper tries to understand the important and common phenomenon of failure modes in PINN training. The local optima hypothesis is questioned with clear evidence by shifting FP32 training to FP64 training, and the network is trainable again.
- The authors deeply dig into the L-BFGS optimizer and provide clear explanations on the reason why FP32 is insufficient for training PINNs.
- The experiments on some commonly used benchmarks of PINNs clearly show that simply shifting the FP32 training to FP64 training resolves the failure modes of vanilla PINNs, without modifications to the model architecture.

Weaknesses:

- The authors could consider apply the FP64 PINN training to some more complex PDEs, like Navier-Stokes equations, to better demonstrate the capability of PINNs with higher precisions.

---

> ### Author Rebuttal · Authors · 2025-07-31
>
> We sincerely appreciate the constructive comments from reviewer QHbM and the time spent on reviewing this paper. We address the questions and clarify the issues accordingly as described below.
>
> >**[Weakness 1]**: The authors could consider apply the FP64 PINN training to some more complex PDEs, like Navier-Stokes equations, to better demonstrate the capability of PINNs with higher precisions.
>
> **[Response to W1]**: We present some experimental results on 2D Naiver-Stoke and Fluid Dynamics in the table below. We find that FP64 consistently improves the performance of the model.
>
>
>
>
> |Problem|Precision|rMAE|rRMSE|
> |---|---|---|---|
> |2D Naiver-Stoke|FP32|13.08|9.08|
> ||FP64|0.0397|0.0285|
> |Fluid Dynamics|FP32|0.3659|0.4082|
> ||FP64|0.1475|0.1721|
>
> The PDE problem are defined as following:
>
> 2D Naiver-Stoke:
>
>
> $\frac{\partial u}{\partial t}+\left(u \frac{\partial u}{\partial x}+v \frac{\partial u}{\partial y}\right)  =-\frac{\partial p}{\partial x}+0.01\left(\frac{\partial^2 u}{\partial x^2}+\frac{\partial^2 u}{\partial y^2}\right)$
>
> $\frac{\partial v}{\partial t}+\left(u \frac{\partial v}{\partial x}+v \frac{\partial v}{\partial y}\right)  =-\frac{\partial p}{\partial y}+0.01\left(\frac{\partial^2 v}{\partial x^2}+\frac{\partial^2 v}{\partial y^2}\right) $
>
> $
> \frac{\partial u}{\partial x}+\frac{\partial v}{\partial y}  =0$
>
> where $u,v$ denotes the velocity on the x and y-axis respectively, and $p$ denotes the pressure field.
>
> Fluid Dynamics:
>
> $\frac{\partial w}{\partial t}+\left(u \frac{\partial w}{\partial x}+v \frac{\partial w}{\partial y}\right)  =\frac{1}{\operatorname{Re}}\left(\frac{\partial^2 w}{\partial x^2}+\frac{\partial^2 w}{\partial y^2}\right), \quad(t, x, y) \in[0, T] \times \Omega $
>
> $ \frac{\partial u}{\partial x}+\frac{\partial v}{\partial y}  =0, \quad(t, x, y) \in[0, T] \times \Omega $
>
> $ w(0, x, y)  =w_0(x, y),(x, y) \in \Omega$
>
> where $u,v$ denotes the velocity  of the fluid on the x and y-axis respectively, $w$ is the vorticity, $T=10$ and $\Omega = [0,2\pi]^2$. $Re = 100$ is Reynolds number, $w_0$ is initial condition.

---

### Official Review · Reviewer_R6jD · 2025-07-02

**Clarity:** 2
**Significance:** 3
**Originality:** 3
**Rating:** 5
**Confidence:** 2

**Summary:**

This paper proposes solving physics informed neural networks in double precision (fp64) instead of the standard float (fp32) or half precision (fp16). Experiments demonstrate that in double precision, many PINNs that typically diverge, instead stably converge to very accurate solves.

**Questions:**

Would it be possible to successfully train PINNs using autocasted fp64 (much like how many torch models are trained using automatic mixed precision between fp32 and fp16/bf16)? E.g. https://docs.pytorch.org/docs/stable/amp.html
While the loss surface analysis is quite interesting, what’s missing is “why” the gradient basins seem so difficult to get underneath / the flatness of the loss landscape. Could the authors display differences in weight matrices in fp32 and fp64 that display what’s going on? Is it just one specific layer or is it all layers, etc.?
Alternatively, is it the imprecision in gradient updates or weight matrices, etc. that are causing the failure modes?

How much slower and less memory efficient is training directly in fp64 instead of fp32? I understand this is most likely 2x the memory and something like 3x slower on the H100s that you used.

Do all of the methods (KAN, PINNMamba, PirateNet, etc..) also work better when run in double precision or is it just PINN in Table 1? I see that there are some clear precision differences in one problem on Figure 8b but don’t really know how that maps.
Given Figure 7b) would a weight normalization or other normalization layer instead improve the training stability of these models?

Is the failure phase in the loss dynamics due to imprecision in the loss function itself? That is, it seems like the model doesn’t generalize well at that point, but is still able to produce very low loss values? What loss function is typically used to train these models?

Are there still cases when fp64 is not enough precision? I know that there are some (mostly Julia unfortunately https://github.com/JuliaMath/DoubleFloats.jl) packages for even higher precision; will these perform even better in MSE? The one missing thing is a bit of a show-stopping experiment – using double allows solving a very difficult scientific PDE that was previously unsolveable in fp32…

Writing / presentation comments:

-	Figure 4: no need to title those plots in matplotlib, and keep the same theme for a/b. Then editorialize the caption some more (also same in Figure 3).

-	Editorialize the caption in Figure 1.

-	L65: “revolutionize” is a bit oversold. Let’s tone that down.

-	L303: “we demystify that the real cause”  “we show that …” and then “This early stopping …”

-	L306: “will enable” - enables

-	L308: honestly, I’m not sure what you’re trying to say here; just delete that sentence.

-	L145: “is then been overthrown” -> rewrite to “[33-36] show that there is near-zero empirical residual loss in many PINN failure modes.”

-	L144: “[20] attributed PINN failures to the “optimizer [getting] stuck … “

-	You probably don’t need both Figures 2 and 6; suggest moving Figure 6 to the front and deleting Figure 2. Then you can editorialize the caption in Figure 2 a lot more.

**Ethical Concerns:**

["NO or VERY MINOR ethics concerns only"]

**Final Justification:**

After rebuttal, I raise score to accept. My questions are answered quite well.

On the H100s, the tensor core fp64 should make things not too slow as compared to fp32. This paper opens an interesting line of research - both practical (how to best run high precisions and emulate them on new gpus) and theoretical (why does gradient underflow occur).

**Quality:**

3

**Strengths And Weaknesses:**

Strengths:
I appreciate the simplicity of this paper – simply running everything in double makes things work better.

I appreciate the loss landscape analysis quite a lot as starts pointing out descriptively out to “why” the optimization fails. Furthermore, I like the “non-existent” loss barrier experiment as it clearly shows that the failure is somehow that of optimization (even the second order optimizers aren’t really able to push through).

I like the difficulty experiment (section 5.3) as well.

Weaknesses:

The paper is a bit too descriptive in nature – there’s not enough “why” experiments pointing out “why” double precision manages to solve things. What should we do when we try this and things still fail to converge (interestingly, we would need harder benchmarks as a result)?
Unfortunately, the experiments are only on one PINN benchmark and not most of them, limiting the generality of the paper.

Figure 5 just shows the optimization landscape becomes very flat for those specific architectures, not that the base PINN also has this issue. Instead showing the PINN loss curve in fp32 and fp64 is probably more interesting.

There's unfortunately a fair number of weird grammatical issues that detract from the meaning of the paper a bit. I've caught a bunch in the questions section, but don't think that's all of them.

---

> ### Author Rebuttal · Authors · 2025-07-31
>
> We sincerely appreciate the constructive comments from reviewer R6jD and the time spent on reviewing this paper. We address the questions and clarify the issues accordingly as described below.
>
>
> >**[Weakness 1]**: The paper is a bit too descriptive in nature – there’s not enough “why” experiments pointing out “why” double precision manages to solve things. What should we do when we try this and things still fail to converge (interestingly, we would need harder benchmarks as a result)? Unfortunately, the experiments are only on one PINN benchmark and not most of them, limiting the generality of the paper.
>
> **[Response to W1]**: We attributed the double-precision problem solving in Section 5.2 to the optimizer's premature determination of model convergence, and we proved our point by analyzing the model's machine epsilon, an analysis whose soundness is backed up by the experiments shown in Figure 7. In further experiments, we determined the variation of computation effective numerical precision with iteration in each benchmark equations. We made some visualization plots, but due to the new rules of NeurIPS rebuttal, we could not present them in the rebuttal stage.  So we do some description here. We found that the computation effective numerical precision of the model will be lower than the machine epsilon after a certain stage of the optimization, accompanied by the gradient of the model being underflowed to 0 in the corresponding case, and thus the weights of the model stop being updated. We guarantee that these new “why” experiments will be presented in camera-ready.
>
> For more challenging equations that are harder to optimise (e.g., a convection equation with an extremely large $\beta$ parameter, such as 200), double precision alone cannot always eliminate the failure modes. Fortunately, our method is orthogonal to all existing remedies for PINN failure modes, so it can be combined with them to further improve the model. We demonstrate the effectiveness of such a combination with a stronger PINN architecture in Section 5.4 and Fig. 8 (b).
>
> On the other hand, for even tougher cases, increasing the arithmetic precision further may resolve the problem; Fig. 8 already shows this trend, and we expect it to generalise to higher precisions. Although mainstream deep-learning frameworks—PyTorch, for example—do not yet support precisions beyond FP64, we are actively working on extending PyTorch and CUDA to higher precisions. Our solution is designed to scale: the new data-type infrastructure can be extended to 128-, 256-bit, or even higher precision. This research is under way, and we plan to release and open-source this support to the community in the near future. We will promptly evaluate the PINN model’s performance at higher numerical precision on more challenging problems and leave this as future work.
>
> In our follow-up work, we ran experiments analogous to those in Figures 4, 5, and 7 on all the additional benchmarks we evaluated. The outcomes were consistent with the conclusions reported in the paper. We will add these results in the next release.
>
>
>
> >**[Weakness 2]**: Figure 5 just shows the optimization landscape becomes very flat for those specific architectures...
>
> **[Response to W2]**: For the loss curves of base PINN, we have presented in Fig. 1, on both FP32 and FP64. The purpose of Fig. 5 is to show what kind of curve a successful optimization of the PINN problem should present.
>
> >**[Weakness 3]**: There's unfortunately a fair number of weird grammatical issues that detract from the meaning of the paper a bit. I've caught a bunch in the questions section, but don't think that's all of them.
>
> **[Response to W3]**: We thank you very much for these comments. We will carefully check the text/grammar word by word before next release. For the captions of individual figures, we will carefully verify and provide self-contained captions.
>
> For example, we will delete the Fig.2 and move Fig. 6 to the front, then add a caption to describe which are failure modes.
>
> >**[Question 1]**: Would it be possible to successfully train PINNs using autocasted fp64 (much like how many torch models are trained using automatic mixed precision between fp32 and fp16/bf16)? While the loss surface analysis is quite interesting, what’s missing is “why” the gradient basins seem so difficult to get underneath / the flatness of the loss landscape. Could the authors display differences in weight matrices in fp32 and fp64 that display what’s going on? Is it just one specific layer or is it all layers, etc.? Alternatively, is it the imprecision in gradient updates or weight matrices, etc. that are causing the failure modes?
>
> **[Response to Q1]**:Unfortunately torch.autocast doesn't support FP64 at the moment. Our team is working on this.
>
> We provide the norms of weights and gradients for FP64 training in Figure 7. We also plot the statistics (mean, std) of the weights and gradients at different numerical precisions. The differences in weights and gradients are reflected on all layers, not some specific layers. These plots will be inserted into the appendix in camera-ready.
>
> Based on the evidence and analysis, we believe that the imprecision in gradient updates is the main reason for the failure modes. Of course, the weights also change due to the imprecision in gradient updates, and the cumulative effect leads to a large difference between the weight matrices of FP32 and FP64. But this difference is not reflected in the norm and statistics of the weights.
>
>
> >**[Question 2]**:How much slower and less memory efficient is training directly in fp64 instead of fp32? I understand this is most likely 2x the memory and something like 3x slower on the H100s that you used.
>
> **[Response to Q2]** We evaluate the FP32 and FP64 training overhead on a Nvidia H100 PCIe GPU, the results are reported in the following table.
>
> |Equation|FP32 rRMSE|FP32 Training Time/Iteration (second)|FP32 Memory (MB)|FP64 rRMSE|FP64 Training Time/Iteration (second)|FP64 Memory (MB)|
> |---|---|---|---|---|---|---|
> |convection| 0.7640 | 0.29 | 1609| 0.0072 |0.32 | 2441|
> |reaction|0.9778|0.26|1629|0.0502|0.29|2481|
> |wave|0.2837|0.47|2295|0.0081|0.61|3845|
> |Allen-Cahn|0.9662|0.40|1975|0.0545|0.48|3167|
>
> By observation, we find that on H100 PCIe, the FP64 latency per iteration is roughly 1.1-1.3 times that of FP32, and the GPU memory footprint is about 1.5-1.7 times that of FP32. This is due to the fact that, with Tensor Core, the FP64 peak arithmetic on H100 PCIe is 60 TFLOPS, the same as the FP32 peak arithmetic. As a result, FP64 is not very much slower than FP32. We observe about 10-30% more latency of FP64 than FP32 that we believe is due to larger memory accesses.
>
> Also, we note that on the Nvidia A6000 GPU, the latency of FP64 is about 10 times that of FP32. This is due to the fact that only few GPU types support high-performance double-precision computation. GPUs known to be suitable for double-precision computing contain the Nvidia A100/A800, H100/H200, and B200.
>
> >**[Question 3]**: Do all of the methods (KAN, PINNMamba, PirateNet, etc..) also work better when run in double precision or is it just PINN in Table 1? ...
>
> **[Response to Q3]** The boost of raising to double precision for the listed baseline model architectures is consistent across these failure mode cases. We provide experimental results on wave equation in the following table. All the models run with double precision.
>
>
> |Model|Loss|rMAE|rRMSE|
> |---|---|---|---|
> |QRes|4.4e-5|0.0092|0.0093|
> |PINNsFormer|4.8e-5|0.0100|0.0102|
> |KAN|3.7e-5|0.0071|0.0072|
> |PirateNet|6.8e-5|0.0135|0.0138|
> |PINNMamba|3.3e-5|0.0062|0.0063|
>
> We also run PINNMamba on FP64 across 4 benchmarks we use.
>
> |Task|Precision|Loss|rMAE|rRMSE|
> |---|---|---|---|---|
> |convection|FP32|0.0001|0.0184|0.0197|
> ||FP64|4e-6|0.0042|0.0047|
> |reaction|FP32|1e-6|0.0092|0.0217|
> ||FP64|1e-6|0.0081|0.0189|
> |wave|FP32|0.0002|0.0193|0.0195|
> ||FP64|4.8e-5|0.0100|0.0102|
> |allen-cahn|FP32|0.0027|0.1432|0.2645|
> ||FP64|9e-6|0.0114|0.0318|
>
> From the above two sets of experiments, we can assume that raising to double precision is an approach with strong generalization capabilities. The results of these experiments, as well as ongoing experiments on all baseline models over all failure mode equations, will be added in camera-ready.
>
> We verified that layer normalization does not work. We speculate that this is due to the fact that the gradient is involved in the computation of the residual loss in PINN, which leads to a distorted mapping of the gradient to the physical information after normalization.
>
> |Method|Training Loss|rMAE|rRMSE|
> |---|---|---|---|
> |Original FP32 Training|0.0133|0.6904|0.7640|
> |Layer Normalization|0.016598|0.7775|0.8394|
>
> >**[Question 4]**: Is the failure phase in the loss dynamics due to imprecision in the loss function itself? ...
>
> **[Response to Q4]** As described in Sec 3.1 and Appendix A, the PINN loss function consists of the physics residual: the network’s predicted solution is substituted into the governing PDE and its initial/boundary conditions, the resulting residual is squared (or taken in absolute value) and summed/averaged over the collocation points, thereby minimizing deviation from the physical laws. The failure modes of PINN appear not only in the inference but also in the training stage, so it is not a generalization problem. This is due to the fact that the training loss of PINN does not directly reflect the absolute error of the solution, but rather the residuals of the governing PDE.
>
> >**[Question 5]**: Are there still cases when fp64 is not enough precision? ...
>
> **[Response to Q5]** Yes, the convection equation with beta=200 can not be solved with PINN with FP64. Unfortunately, these computational libraries for fp128 only support cpu, and as mentioned in response to W1, we are developing Pytorch support for higher precision. We see Allen-Cahn as a problem that FP32 failed to solve, but worked wonderfully well with FP64.

---

> > ### Comment · Reviewer_R6jD · 2025-08-04
> >
> > Thanks for the comments, I've increased my score to accept.
> >
> > > Unfortunately, these computational libraries for fp128 only support cpu, and as mentioned in response to W1, we are developing Pytorch support for higher precision.
> >
> > In general, emulating the higher precisions (even fp64) is a worthwhile challenge on H100s and other newer gpus, with broader impact beyond just this work.
> >
> > > with Tensor Core, the FP64 peak arithmetic on H100 PCIe is 60 TFLOPS, the same as the FP32 peak arithmetic. As a result, FP64 is not very much slower than FP32. We observe about 10-30% more latency of FP64 than FP32 that we believe is due to larger memory accesses.
> >
> > This is a very good point; the tensor core arithmetic should reduce slowdowns.

---

> > > ### Author Response · Authors · 2025-08-04
> > >
> > > Thank you for your response. We appreciate your support in the acceptance of our paper. If you have any further concerns and questions, we are willing to discuss them with you.
> > >
> > > Authors

---

### Official Review · Reviewer_y1Ny · 2025-07-03

**Clarity:** 3
**Significance:** 3
**Originality:** 3
**Rating:** 4
**Confidence:** 5

**Summary:**

The paper analyzes the failure modes of training physics-informed neural networks. It finds that the reason for failure in the loss function is precision limitation rather than getting stuck in local minima during optimization. To achieve this new insight, the authors propose the same-basin hypothesis of the loss landscape, where the correct values exist in the same basin, but since the loss landscape becomes flat due to precision limitations, optimization comes to a halt. This hypothesis is supported by analyses of physics-informed neural network training with various initializations and the absence of loss barriers from failure mode to optimization approach. To address this issue, the paper suggests using FP64 instead of FP32. Moreover, the paper demonstrates the efficacy of FP64 in solving more challenging equations with physics-informed neural networks.

**Questions:**

Why does the PINNMamba model in the reaction benchmark show better performance compared to the PINN_FP64 model?

**Ethical Concerns:**

["NO or VERY MINOR ethics concerns only"]

**Final Justification:**

The authors evaluate the overhead of FP64 training compared to FP32 training on various GPUs and show that the overhead is negligible. Various quantization approaches, such as using scaling factors and stochastic rounding, are also evaluated, but have shown limited progress. Although better quantization approaches might be found for this problem, the FP64 solution is sufficient as an introductory solution and opens valuable research directions for the future.

**Limitations:**

yes

**Paper Formatting Concerns:**

There are no concerns regarding paper formatting.

**Quality:**

3

**Strengths And Weaknesses:**

Strength:

The analysis of the loss landscape of physics-informed neural networks and the same-basin hypothesis is interesting and novel.

The paper is well-written and well-organized.

Weaknesses:

1- The overhead of using FP64 for training physics-informed neural networks is not explored quantitatively. For example, this overhead could be evaluated on a simple model that works with both FP32 and FP64 on GPU.

2- The FP64 solution proposed by the author is a rudimentary approach and moves in the opposite direction of optimizing neural networks toward low-precision arithmetic. The author should analyze the gradients, weights, and activation distributions more thoroughly and demonstrate whether simple approaches, such as FP32 with scaling factors or grouping parameters, work or not. The author can also draw inspiration from many approaches used for FP16 and BF16 training such as stochastic rounding and apply them to FP32 for PINNs.

---

> ### Author Rebuttal · Authors · 2025-07-30
>
> We sincerely appreciate the constructive comments from reviewer y1Ny and the time spent on reviewing this paper. We address the questions and clarify the issues accordingly as described below.
>
> >**[Weakness 1]**:  The overhead of using FP64 for training physics-informed neural networks is not explored quantitatively. For example, this overhead could be evaluated on a simple model that works with both FP32 and FP64 on GPU.
>
> **[Response to W1]**: Thank you for your comments. We evaluate the FP32 and FP64 training overhead on a Nvidia H100 PCIe GPU, the results are reported in the following table.
>
> |Equation|FP32 rRMSE|FP32 Training Time/Iteration (second)|FP32 Memory (MB)|FP64 rRMSE|FP64 Training Time/Iteration (second)|FP64 Memory (MB)|
> |---|---|---|---|---|---|---|
> |convection| 0.7640 | 0.29 | 1609| 0.0072 |0.32 | 2441|
> |reaction|0.9778|0.26|1629|0.0502|0.29|2481|
> |wave|0.2837|0.47|2295|0.0081|0.61|3845|
> |Allen-Cahn|0.9662|0.40|1975|0.0545|0.48|3167|
>
> By observation, we find that on H100 PCIe, the FP64 latency per iteration is roughly 1.1-1.3 times that of FP32, and the GPU memory footprint is about 1.5-1.7 times that of FP32. This is due to the fact that, with Tensor Core, the FP64 peak arithmetic on H100 PCIe is 60 TFLOPS, the same as the FP32 peak arithmetic. As a result, FP64 is not very much slower than FP32. We observe about 10-30% more latency of FP64 than FP32 that we believe is due to larger memory accesses.
>
> Also, we note that on the Nvidia A6000 GPU, the latency of FP64 is about 10 times that of FP32. This is due to the fact that only few GPU types support high-performance double-precision computation. GPUs known to be suitable for double-precision computing contain the Nvidia A100/A800, H100/H200, and B200.
>
> >**[Weakness 2]**: The FP64 solution proposed by the author is a rudimentary approach and moves in the opposite direction of optimizing neural networks toward low-precision arithmetic. The author should analyze the gradients, weights, and activation distributions more thoroughly and demonstrate whether simple approaches, such as FP32 with scaling factors or grouping parameters, work or not. The author can also draw inspiration from many approaches used for FP16 and BF16 training such as stochastic rounding and apply them to FP32 for PINNs.
>
> **[Response to W2]**: We agree that the current trend in AI is towards efficient low-precision computation. The development of Quantization techniques has played an important role in areas such as NLP, CV, and especially LLM. But with this paper we would like to draw the attention of the AI community to the possible role of numerical precision. As an example, Pytorch defaulted to matmul.allow_tf32 = True in version 1.7, which is feasible for general AI computational problems, but fatal for scientific computation.
>
> We did an analysis of the gradient, weights for the PINN model under multiple precision runs. We plotted visualizations like Fig. 7(b),(c) for the variation of statistics such as the norm, mean, and variance of the gradients and weights by iteration. Unfortunately, due to the new NeurIPS rebuttal rules this year, we cannot provide these visualizations at the rebuttal stage. We will put up these visualizations in the next release. What we have found in these visualizations is that at low precision, the gradient goes to zero prematurely, resulting in the weights not changing, which is consistent with our analysis.
>
>
> We take the convection equation as an example and append some common ways to change the scale of weight and gradient as shown in the table below. The model is trained with FP32 precision. We find that layer normalization, simply stochastic rounding from FP32 to FP64, and scaling factors do not inherently improve the failure modes of PINN. We believe that the core reason for the failure of these methods is that the gradients on spacetime are involved in the computation of residual loss in PINN, which leads to the failure of the scaling methods in terms of the higher-order derivatives taken when the gradients continue to be solved for these differentials. The failure of the simple quantization approach that we tested indicates that for the successful operation of PINNs at low precision, it is still a new area to explore and a new research direction.
>
> |Method|Training Loss|rMAE|rRMSE|
> |---|---|---|---|
> |Original FP32 Training|0.0133|0.6904|0.7640|
> |Layer Normalization|0.016598|0.7775|0.8394|
> |Scaling Factors (1e7)|0.014238|0.7431|0.8125|
> |Stochastic Rounding|0.013105|0.7178|0.7903|
> |FP64 Training|5e-6|0.0059|0.0072|
>
>
>
>
> >**[Question 1]**: Why does the PINNMamba model in the reaction benchmark show better performance compared to the PINN_FP64 model?
>
> **[Response to Q1]**: A core contribution of our paper is that we understand PINN failure modes from the perspective of PINN optimization. Our theory illustrates that the nature of PINN failure modes is an optimization problem in the flat plane on the loss landscape. Training with FP64 is one perspective to improve optimization, meanwhile, good model architecture, good sampling method, good optimizer, etc. are all ways to improve optimization. Although there is a slight numerical difference, both FP64 and PINNMamba solve the failure modes of PINN, while our method is more straightforward and has stronger generalization ability than PINNMamba. PINNMamba has a more optimizable model architecture than PINN, and we illustrate in Fig. 8 (b) the combined effectiveness of PINNMamba ( as well as other model structures) and FP64, which are two independent perspectives to improve the optimization of PINN, and can be used in conjunction with each other. Our main thrust in this paper is to make the AI-based scientific computing community more aware of the impact of numerical precision.

---

> > ### Comment · Reviewer_y1Ny · 2025-08-05
> >
> > Thank you for the complete response. My questions were answered well. Although there might be a better solution than using  FP64 (which increases overhead slightly), the paper introduces a new idea and a valuable research direction. I have revised my score to 4.

---

### Decision · Program_Chairs · 2025-09-17

**Decision:**

Accept (poster)

**Comment:**

This paper argues that common “failure modes” in Physics-Informed Neural Networks (PINNs) stem not from local minima but from insufficient precision. With FP32, L-BFGS prematurely convergences, causing spurious failure, while FP64 removes these stalls and enables reliable PDE solving. The authors identify a three-stage training dynamic (un-converged, failure, success) governed by arithmetic precision, reframing failures as precision-induced and highlighting the need for high-precision computation.

The reviewers consistently find the paper well-motivated and easy to follow. They highlight the novelty of the loss landscape analysis, the practical simplicity of the proposed solution, and the thorough evaluation showing that failures can be resolved without architectural changes. However, they also raise concerns. The approach is seen as somewhat rudimentary, with insufficient analysis of why FP64 works, no quantitative assessment of its computational cost, and limited exploration of alternative precision strategies. The experiments are viewed as narrow in scope—restricted to simple PDEs, L-BFGS optimization, and a single benchmark—leaving questions about generality, applicability to harder PDEs, and effectiveness with other optimizers like Adam. Issues with clarity of writing and figures are also noted.

Based on the strengths and weaknesses discussed, the AC believes the strengths outweigh the weaknesses and recommends acceptance, while strongly encourages the authors to incorporate the rebuttal into the final version of the paper.